# Quality of Life in COVID-Related ARDS Patients One Year after Intensive Care Discharge (Odissea Study): A Multicenter Observational Study

**DOI:** 10.3390/jcm12031058

**Published:** 2023-01-29

**Authors:** Cristian Deana, Luigi Vetrugno, Andrea Cortegiani, Silvia Mongodi, Giulia Salve, Matteo Mangiagalli, Annalisa Boscolo, Tommaso Pettenuzzo, Sara Miori, Andrea Sanna, Sergio Lassola, Sandra Magnoni, Elena Ferrari, Emanuela Biagioni, Flavio Bassi, Nadia Castaldo, Alberto Fantin, Federico Longhini, Francesco Corradi, Francesco Forfori, Gianmaria Cammarota, Edoardo De Robertis, Danilo Buonsenso, Savino Spadaro, Domenico Luca Grieco, Maria De Martino, Miriam Isola, Francesco Mojoli, Massimo Girardis, Antonino Giarratano, Elena Giovanna Bignami, Paolo Navalesi, Maurizio Cecconi, Salvatore Maurizio Maggiore

**Affiliations:** 1Department of Anesthesia and Intensive Care, Health Integrated Agency of Friuli Centrale, 33100 Udine, Italy; 2Department of Medical, Oral and Biotechnological Sciences, University of Chieti-Pescara, 66100 Chieti, Italy; 3Department of Anesthesiology, Critical Care Medicine and Emergency, SS. Annunziata Hospital, 66100 Chieti, Italy; 4Department of Surgical, Oncological and Oral Science (DiChirOnS), University of Palermo, 90127 Palermo, Italy; 5Anesthesia and Intensive Care, Fondazione IRCCS Policlinico S. Matteo, 27100 Pavia, Italy; 6Unit of Anesthesia and Intensive Care, Department of Clinical-Surgical, Diagnostic and Pediatric Sciences, University of Pavia, 27100 Pavia, Italy; 7Soleterre, Strategie di Pace ONLUS, 20123 Milan, Italy; 8Institute of Anaesthesia and Intensive Care, Padua University Hospital, 35128 Padua, Italy; 9Department of Medicine (DIMED), University of Padua, 35128 Padua, Italy; 10Anesthesia and Intensive Care 1, Santa Chiara Hospital, 38122 Trento, Italy; 11Intensive Care Unit, University Hospital of Modena, University of Modena Reggio Emilia, 41124 Modena, Italy; 12Pulmonology Unit, Health Integrated Agency of Friuli Centrale, Academic Hospital of Udine, 33100 Udine, Italy; 13Anesthesia and Intensive Care Unit, Department of Medical and Surgical Sciences, University Hospital Mater, Domini, Magna Graecia University, 88100 Catanzaro, Italy; 14Department of Surgical, Medical and Molecular Pathology and Critical Care Medicine, University of Pisa, 56126 Pisa, Italy; 15Department of Medicine and Surgery, University of Perugia, 06121 Perugia, Italy; 16Department of Woman and Child Health and Public Health, Fondazione Policlinico Universitario A. Gemelli IRCCS, 00168 Rome, Italy; 17Centro di Salute Globale, Università Cattolica del Sacro Cuore, 00168 Rome, Italy; 18Anesthesiology and Intensive Care, Department of Translational Medicine, Faculty of Medicine and Surgery, University of Ferrara, 44121 Ferrara, Italy; 19Department of Anesthesiology and Intensive Care Medicine, Catholic University of The Sacred Heart, 00168 Rome, Italy; 20Department of Anesthesia, Emergency and Intensive Care Medicine, Fondazione Policlinico Universitario A. Gemelli IRCCS, 00168 Rome, Italy; 21Department of Medicine, University of Udine, 33100 Udine, Italy; 22Anesthesiology, Critical Care and Pain Medicine Division, Department of Medicine and Surgery, University of Parma, 43126 Parma, Italy; 23Department of Biomedical Sciences, Humanitas University, 20072 Milan, Italy; 24IRCCS Humanitas Research Hospital, Rozzano, 20089 Milan, Italy; 25Department of Innovative Technologies in Medicine and Dentistry, Gabriele d’Annunzio University of Chieti Pescara, 66100 Chieti, Italy

**Keywords:** COVID-19, health related quality of life, post-traumatic stress disorder, ICU, ARDS, SF-36, impact of event scale-revised

## Abstract

Background: Investigating the health-related quality of life (HRQoL) after intensive care unit (ICU) discharge is necessary to identify possible modifiable risk factors. The primary aim of this study was to investigate the HRQoL in COVID-19 critically ill patients one year after ICU discharge. Methods: In this multicenter prospective observational study, COVID-19 patients admitted to nine ICUs from 1 March 2020 to 28 February 2021 in Italy were enrolled. One year after ICU discharge, patients were required to fill in short-form health survey 36 (SF-36) and impact of event-revised (IES-R) questionnaire. A multivariate linear or logistic regression analysis to search for factors associated with a lower HRQoL and post-traumatic stress disorded (PTSD) were carried out, respectively. Results: Among 1003 patients screened, 343 (median age 63 years [57–70]) were enrolled. Mechanical ventilation lasted for a median of 10 days [2–20]. Physical functioning (PF 85 [60–95]), physical role (PR 75 [0–100]), emotional role (RE 100 [33–100]), bodily pain (BP 77.5 [45–100]), social functioning (SF 75 [50–100]), general health (GH 55 [35–72]), vitality (VT 55 [40–70]), mental health (MH 68 [52–84]) and health change (HC 50 [25–75]) describe the SF-36 items. A median physical component summary (PCS) and mental component summary (MCS) scores were 45.9 (36.5–53.5) and 51.7 (48.8–54.3), respectively, considering 50 as the normal value of the healthy general population. In all, 109 patients (31.8%) tested positive for post-traumatic stress disorder, also reporting a significantly worse HRQoL in all SF-36 domains. The female gender, history of cardiovascular disease, liver disease and length of hospital stay negatively affected the HRQoL. Weight at follow-up was a risk factor for PTSD (OR 1.02, *p* = 0.03). Conclusions: The HRQoL in COVID-19 ARDS (C-ARDS) patients was reduced regarding the PCS, while the median MCS value was slightly above normal. Some risk factors for a lower HRQoL have been identified, the presence of PTSD is one of them. Further research is warranted to better identify the possible factors affecting the HRQoL in C-ARDS.

## 1. Introduction

Since the beginning of the pandemic, more than half a billion people worldwide have suffered from the novel coronavirus disease (COVID-19) [1].

Patients with severe acute hypoxemic respiratory failure have several risk factors for post-intensive care syndrome (PICS), including prolonged invasive mechanical ventilation, development of intensive care unit (ICU)-acquired weakness, steroids and neuromuscular blocking agent administration, which has been demonstrated to reduce the health-related quality of life (HRQoL) after ICU discharge [2,3,4].

Herridge et al. described a persistently reduced HRQoL that lasted several months to years in patients who suffered from acute respiratory distress syndrome (ARDS) [5]. Other studies confirmed their findings and further demonstrated that critically ill patients who survived ICU discharge presented persistent physical and mental impairment [6,7,8]. 

However, how the same factors affect COVID-19 ARDS (C-ARDS) patients is still poorly known. Limited resources during the first phase of the pandemic, coupled with ineffective treatments and insufficient evidence of C-ARDS management, may have led to increased mortality and worse recovery from critical COVID-19 illness [9]. 

Preliminary studies in C-ARDS patients at 3- and 6-month follow-up revealed an impaired HRQoL after ICU discharge [10,11]. Following this period, improvements in physical symptoms and performance have been reported in studies in a relatively small cohort of patients [12]. Thus, the quality of life in C-ARDS survivors at longer follow-up remains poorly investigated. 

The primary aim of this study was to evaluate the HRQoL after 1 year of ICU discharge in COVID-19 survivors.

Secondary aims included identifying possible correlations between HRQoL 1 year after ICU discharge and the demographic, medical or clinical data (during hospitalization), screening for post-traumatic stress disorder (PTSD), detecting the risk factors and investigating whether PTSD has an impact on the HRQoL.

## 2. Materials and Methods

### 2.1. Study Setting and Design 

A prospective multicenter observational study was conducted in nine Italian ICUs, eight academic and one nonacademic, after the Ethics Committee of Friuli Venezia Giulia Region (Udine, Italy), as the coordinating center (CEUR-2021-Os-99), approved the study. 

Principal investigators for each location were responsible for obtaining the local ethics committees’ approval and patients’ consent to participate in the study. This was completed following hospital protocols and institutional regulations during the COVID-19 emergency. This study was prospectively registered at ClinicalTrials.gov number: NCT04860687 (registered on 27th April 2021). This work follows the STROBE checklist.

### 2.2. Patients’ Characteristics

The patients included in the study had a positive COVID-19 assay from either nasal or pharyngeal swabs or lower respiratory tract aspirates, and were admitted to ICU due to acute hypoxemic respiratory failure and survived the ICU stay.

Excluded patients who had known cognitive disorders (medical history positive for dementia, delirium or loss of memory previous to ICU admission), psychological disorders (depression, history of previous PTSD), advanced malignancies (under palliative care), end-stage organ disease at ICU admission (defined as patient requiring chronic hemodialysis, mechanical cardiocirculatory support, palliative care to resolve symptoms related to cardiac disease refractory to all available treatments, long-term oxygen therapy, on a waiting list for solid organ transplantation or with cirrhosis but excluded from liver transplantation) or did not require mechanical ventilatory support during their ICU stay. 

### 2.3. Outcomes

The quality of life after ICU discharge was assessed with the short-form health survey-36 (SF-36), while PTSD was screened with the impact of event scale-revised (IES-R).

The SF-36 is a 36-item patient-reported questionnaire that evaluates the HRQoL. The SF-36 produces eight scaled scores that are the weighted sums of the questions in their section. Each scale is directly transformed into a 0–100 scale on the assumption that each question carries equal weight. The higher the score, the better the quality of life and vice versa. 

The domains of the SF-36 are physical functioning (PF), which reflects the extent to which general health limits physical activity; physical role (PR), which expresses how physical health interferes with work or limits activity; bodily pain (BP), which analyzes the intensity of pain and the effect of pain on a patient’s ability to work; general health (GH), a patient’s own evaluation of his or her health or health outlook; vitality (VT), which includes the energy the patient has; social functioning (SF), a measure of how health or emotional problems interfere with social activities; emotional role (RE), an evaluation of the extent to which emotional problems interfere with work or activities; and mental health (MH), a global assessment of general mental health. The SF-36 can be filled in by the patient alone or with the help of relatives. Moreover, it can be administered by phone.

The eight SF-36 domains could be collapsed to create two global components: the physical component summary (PCS) and mental component summary (MCS), according to the method proposed by Ware et al. [13].

The PCS is principally derived from PF, PR and BP, while for the MCS, major determinants are MH, RE and SF. VT and GH are equally determinants of both summary scores. In practical, the PCS reflects the physical wellness, while MCS returns information on the global mental health condition.

They are constructed using a principal component analysis, based on the data of the general population of the US, standardized to obtain a mean of 50 and a standard deviation of 10.

The IES-R is a 22-item questionnaire that measures the subjective distress caused by traumatic events. It is a self-reported scale, with items rated on a 5-point Likert scale ranging from 0 to 4, with a minimum of 0 to a maximum total score of 88. There are also three subscale scores that define intrusion, avoidance and hyperarousal aspects of PTSD. Sum scores equal to or greater than 33 or a mean cutoff value equal to or greater than 1.75 for overall questions indicate the probable presence of PTSD, as described in Appendix A [14].

### 2.4. Data Collection

Local investigators were responsible for screening and determining the patients’ inclusion, specifically considering a 1-year follow-up after ICU discharge. The same investigators also contacted patients to complete the self-reported short-form health survey-36 (SF-36) questionnaire to analyze the HRQoL and IES-R, as a screening tool for PTSD.

Patients were contacted by phone and asked to complete the questionnaires independently or with the help of a relative. Results were delivered either by mail or e-mail depending on individual’s preference. Patients who did not respond after three phone calls were considered unavailable and excluded from the study; those who agreed to participate but did not send the completed questionnaires, despite three reminders, were considered lost to follow-up. 

The following demographic data were recorded: age, gender, weight at ICU admission and at follow-up, APACHE II score, lung injury score (LIS score) [15], level of education (lower education: <8 years of school [compulsory school], higher education >8 years of school [high school degree or college degree]), marital status (single, married, separated/divorced, widowed), and employment (jobless, active worker, retired), previous medical history of cardiac, pulmonary, kidney or liver disease, diabetes, time between hospital admission and ICU admission (days), length of ICU stay (LOS_ICU_), length of hospital stay (LOS_HOSP_), and ward of destination after ICU discharge. 

Clinical ICU data included the type of mechanical ventilation (noninvasive vs. invasive), use of steroids and neuromuscular blocking agents and their respective duration, need for renal replacement therapy, duration of mechanical ventilation and the eventual need for tracheostomy. 

### 2.5. Statistical Analysis

Categorical variables were presented as absolute values (percentages) and continuous variables as medians and interquartile ranges [IQRs]. Normality was assessed using the Shapiro–Wilk test. Categorical variables were compared using the chi-square test or Fisher’s exact test, as appropriate. Univariable and multivariable linear regressions were performed to estimate the associations between the SF-36 domains, IES-R and the clinical/demographic variables by calculating β (linear regression coefficient) and 95% confidence intervals (CIs). Univariable and multivariable logistic regressions were performed to explore variables associated with the presence of PTSD by estimating the odds ratios (OR, 95% CI). Multivariable analyses included all significant variables, *p* < 0.05 from univariable analyses, and considered potential collinearities. Results were adjusted for each hospital center. No imputation was carried out for missing data. Statistical analyses were performed using STATA 17.

### 2.6. Sample Size

According to Herridge’s study, surviving ARDS patients showed a median PF of 60 (IQR 35–85) 1 year after ICU discharge [5]. Given this value, Wan’s method was used to obtain a mean value of 60 and standard deviation of 37 [16]. A convenience sample size of 340 patients produces a two-sided 95% confidence interval for this mean with a precision of 4%, an α level of 0.05. Physical functioning was chosen because this domain asks respondents to report limitations on 10 mobility activities, such as walking specified distances, carrying groceries, bathing or dressing. This fully reflects the extent to which general health, also in patients experiencing PICS, limits daily life physical activity, with a consequent impact on the HRQoL.

## 3. Results

In all, 1003 patients with C-ARDS were admitted to the nine participating ICUs from 1 March 2020 to 28 February 2021. The follow-up ended on 9 April 2022 given that the last patient included was discharged from ICU on 9 April 2021. The final statistical analysis included 343 patients who satisfied the inclusion and exclusion criteria, as shown in the study flow chart (Figure 1). 

The median age was 63 years. The majority of patients were men (79.3%). They required critical care admission in the absence of previously known pulmonary conditions. Approximately half of the patients had arterial hypertension (42.9%), and 14.2% had diabetes. At 6.1%, COPD was the most represented pulmonary comorbidity. In all, 54% of patients had a high level of education and 79.6% were married or cohabiting at the time of the study. In contrast, 58 (17%) patients lived alone. In all, 159 (46.3%) patients were active workers, and 45.5% were retired (Table 1). 

Median APACHE II and LIS scores at ICU admission were, respectively, 11 [8–14] and 3 [2.3–3]. In all, 91% of patients required endotracheal intubation and invasive mechanical ventilation (MV) with a median duration of 10 days [2–20]. Neuromuscular blockers were administered to 111 (32.3%) patients for a median of 96 h [IQR 36–160]. In all, 194 (56.5%) patients received steroids for 10 days [7–13]. Some patients (6.7%) needed renal replacement therapy, while 87 (25.4%) were tracheostomized during their ICU stay. Median LOS_ICU_ was 13 days [IQR 6–25], and 26.5% of patients were discharged to a high dependency unit (HDU), 24% to pulmonology and 24.1% to a medical ward. Median LOS_HOSP_ was 32 days [IQR 19–47] (Table 2). 

The HRQoL at 1-year follow-up of C-ARDS patients is shown in Table 3.

The median physical component score and mental component score were 45.9 (36.5–53.5) and 51.7 (48.8–54.3), respectively, as shown in Figure 2.

In all, 109 (31.8%) tested positive for PTSD, as defined by a IES-R sum score ≥ 33. In this group, the median value for avoidance was 1.9 [IQR 1.1–2.3] and 2.4 [IQR 1.9–2.3] for intrusion, while hyperarousal had a median score of 2 [IQR 1.3–2.7]. The median sum score was 47 [IQR 40–57]. When comparing the HRQoL in COVID-19 survivors, all SF-36 domains scores were significantly higher in the group that did not develop PTSD (Table 3 and Figure 3). 

The box plot shows the median and interquartile ranges, while the whiskers represent the outliers. The red dotted line represents the value for the normal population.

To explore the possible factors related to the HRQoL, a multivariate linear regression analysis was performed and revealed that the female gender (β = −6.97, *p* = 0.044), a history of cardiovascular (β = −10.02, *p* = 0.001), liver disease (β = −17.07, *p* = 0.036) and LOS_HOSP_ (β = −0.21, *p* = 0.001), reduced the PF score.

LOS_HOSP_ also influenced the PR score (β = −0.31, *p* = 0.01). Weight at baseline (β = −0.31, *p* = 0.017) and a history of pulmonary disease (β = −12.51, *p* = 0.048) significantly influenced the RE score. The BP was lower in patients with known pulmonary disease (β = −15.76, *p* < 0.001). A history of cardiovascular and pulmonary disease also reduced the GH score, respectively, at β = −6.04 (*p* = 0.022) and β = −8.47 (*p* = 0.017). The female gender was associated with a lower VT score (β = −8.10, *p* = 0.005). The SF was lower in patients with cardiovascular disease, β = −6.58 (*p* = 0.035), and with a longer LOS_HOSP_, β = −0.15 (*p* = 0.042). Lastly, regarding the SF-36, health change was higher in patients who received NMB during their ICU stay (β = 8.67, *p* = 0.040). Complete multivariable analyses for each SF-36 domain are provided as Appendix A. 

In addition, the risk factors for lower PCS and MCS scores were investigated, as reported in Appendix A. LOS_HOSP_ negatively affected the PCS (β = −0.07, *p* = 0.007).

An evaluation of the possible factors related to PTSD was also performed. Weight at follow-up was found to be a risk factor for PTSD, OR 1.02 [IQR 1.00–1.04], *p* = 0.03.

Other risk factors for PTSD are reported in Appendix A.

## 4. Discussion

This multicenter observational study reports C-ARDS patients’ HRQoL 1 year after ICU discharge including the identification of clinical/demographic factors that correlate with it. The physical component summary was lower than normal. The mental component summary was slightly above the reference value for the normal population.

Risk factors for a lower PF score, which is a major determinant of the PCS, were the female gender, a history of cardiovascular or liver disease and LOS_HOSP_. The latter factor negatively also affected the PCS. Nearly one out of three patients tested positive for PTSD and had a significantly lower HRQoL.

Eberst et al. found results akin to ours with C-ARDS patients, showing higher HRQoL scores than those in Herridge’s report [17]. 

In the multicenter study of 118 Dutch C-ARDS survivors, Vlake et al. described comparable results, although the HRQoL was superior to their traditional cohort of critically ill patients [13]. In contrast with our work, Eberst’s was a single center study with a small sample size, and Vlake et al. limited the follow-up to 6 months after ICU discharge. 

The literature includes frequent reports of a reduced HRQoL after traditional ARDS due to the physical impairment that sometimes requires years of external support [18,19,20]. This has an impact on patients’ families as well as society. Marti et al., in addition to the HRQoL after 1 year of ICU discharge, investigated the economic costs of traditional ARDS survivors. They advocated further research to optimize resources and improve outcomes because they discovered that patient care had high costs and, in most cases, the HRQoL remained low [21]. 

There is a substantial difference between traditional ARDS and C-ARDS populations: C-ARDS patients were mainly men, with severe acute pulmonary disease and a high demand for invasive mechanical ventilation (91%), but unlike traditional ARDS, they mainly suffered from acute single organ failure at ICU admission. In fact, considering that sepsis with multi-organ involvement was the main cause of ICU admission for traditional ARDS [17], in COVID-19 critically ill patients with lung damage was the main feature at ICU admission. Then, with the ongoing and massive activation of the immune system, the single organ failure often became a multiple organ dysfunction (MODS-CoV-2) [22,23].

In fact, brain, heart, liver and coagulation disorders have been frequently reported as the consequence of MODS-CoV-2 [24,25,26,27]

Further complications were caused by tight triage since demand for ICU admission increased as infection rates peaked in a context of limited resources [28,29]. This probably resulted in ICU admission being denied for patients with high predicted mortality, favoring better outcomes for those granted ICU admission [30,31,32,33].

We found that the female gender negatively influenced the PF (β = −7.13). This result aligns with a recent study by Huang et al. [11] in which females had persistent symptoms while also confirming findings by Brown et al. in which female ICU survivors, after traditional ARDS, were associated with a lower HRQoL at follow-up [34]. This should translate into the need for strict follow-up after ICU discharge in women who suffered from C-ARDS.

A history of cardiovascular disease was also strongly correlated with a worse PF at follow-up (β = −10.4, *p* = 0.001). Considering that the most represented cardiovascular disease in this cohort was arterial hypertension, this correlation sheds new light on the role of hypertension in COVID-19 outcomes [35,36]. 

It is well known that arterial hypertension creates endothelial dysfunction, and numerous studies have also demonstrated that hypertension per se reduces the HRQoL [37]. SARS-CoV-2 infection leads to persistent endothelial dysfunction and thus increases the risk of developing “long COVID” (persistence of physical symptoms) [38,39]. Consequently, we hypothesize that endothelial dysfunction in hypertensive COVID-19 patients could play a major role in the HRQoL reduction. 

Duration of hospital stay was related to a lower PF at follow-up (β = −0.22, *p* = 0.001), and in general with a reduced PCS (β = −0.07, *p* = 0.007). Long hospital stay is also a well-established risk factor for PICS that could last for months after ICU discharge impairing the quality of life [40]. 

Nevertheless, prolonged recovery from acute illness often implies longer hospitalization in general and in rehabilitation wards. LOS_HOSP_ in this study was not different from that of other studies involving severe ARDS patients [21].

An interesting finding is that 31.8% of patients screened with IES-R had PTSD. This was tested in C-ARDS patients at long-term follow-up after ICU discharge. Previous coronavirus epidemics, severe acute respiratory syndrome coronavirus and Middle East respiratory syndrome coronavirus, highlighted the increased psychologic distress with PTSD, depression and anxiety [41,42]. 

A recent metanalysis of non-COVID patients found that 12 months after ICU discharge, 19.8% presented PTSD [43], which is a lower result than this study. However, the first studies on PTSD after C-ARDS seem to confirm that its incidence is about 30% [44,45,46]. Furthermore, the high burden of PTSD might have been the result of the near complete denial in ABCDEF bundles during the peak of contagions [47].

Additional risk factors for PTSD development include social limitations and restrictions, media overpressure, restricted hospital visits and protective clothes and masks that hindered face-to-face interactions between patients and health-care workers [48,49]. Moreover, infectious disease survivors, including those who survived SARS-CoV-2, are exposed to psychological risks due to public fear of the disease and contagiousness that may result in extensive isolation [50]. Curiously, experimental studies found that PTSD could be an expression of endothelial dysfunction after viral infection, a well-represented feature in COVID-19 patients [51]. 

Given that we reported a significantly better HRQoL in the subgroup of C-ARDS patients without PTSD, a search for PTSD risk factors was performed. A multivariate analysis revealed that a higher weight at follow-up was a risk factor for PTSD (OR 1.02). Similar results were obtained by Tarsitani et al. when they evaluated a mixed cohort of critical and noncritical COVID patients. They advocated further investigations to determine if obesity could be a risk factor for PTSD or vice versa [52] because evidence in non-COVID-19 patients highlighted that PTSD could result in faster weight gain [53]. Therefore, it is possible to speculate that increased weight is probably a consequence of PTSD rather than a risk factor.

The literature has yet to fully assess the risks and protective factors that determine mental health outcomes after COVID-19 [54,55,56,57,58,59]. To the best of our knowledge, no studies have evaluated the impact of employment on the incidence of PTSD after C-ARDS. 

There are some limitations to this study that include the absence of the SF-36 and IES-R baseline values. However, the patients were young and presented few comorbidities at baseline, so it is possible to speculate that they should have no reasons for the low SF-36 scores at baseline. In addition, patients with a previous history of cognitive and psychological disorders were excluded. 

Some missing patients that did not accept to participate or were lost to follow-up could have had a poor performance status after ICU discharge, limiting their ability to answer the questionnaires. However, we cannot exclude that some of them probably were less sick and felt to be in a good health status that they thought they had nothing to report. It is our opinion that these opposite behaviors may limit the selection bias.

Patients who tested positive for PTSD through the IES-R were not further evaluated with psychiatric consultation, but restricted access to hospitals in the acute phase of the pandemic has limited this evaluation. Our findings reflect the characteristics of our population, hence our results must be taken cautiously. 

Finally, patients performed a self-reported questionnaire that could not be collinear with test objectivation, such as a 6-min walking test. In fact, as Latronico et al. suggested, objective tests may reveal functional impairment, regardless of good scores on self-reported questionnaires [60].

## 5. Conclusions

The HRQoL in C-ARDS patients was reduced in the PCS, while the PCS was quite maintained in the overall population. More than 30% of patients tested positive for PTSD. In this group, the HRQoL was significantly lower than in those not reporting PTSD at follow-up. Continuous and prolonged follow-up of C-ARDS survivors, the HRQoL evaluation and PTSD screening are highly advocated because evidence is still low. It is imperative to obtain better outcomes and a better HRQoL, especially during a pandemic in which a large part of the population required hospitalization.

## Figures and Tables

**Figure 1 jcm-12-01058-f001:**
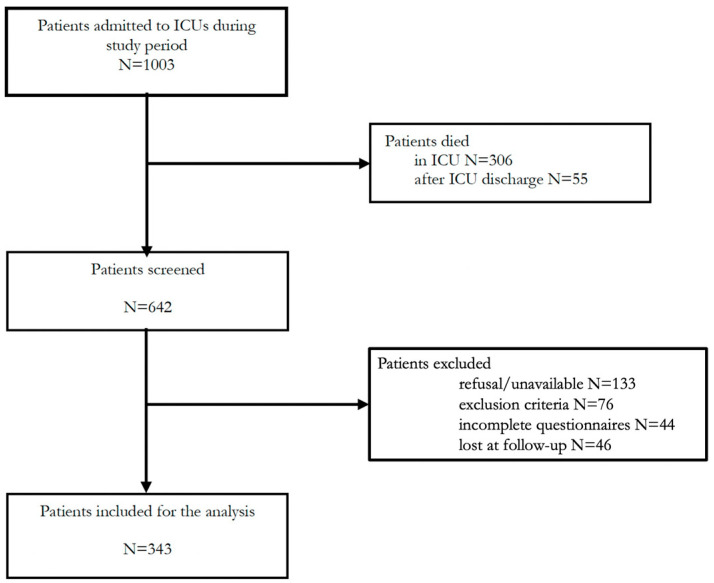
Study flow chart.

**Figure 2 jcm-12-01058-f002:**
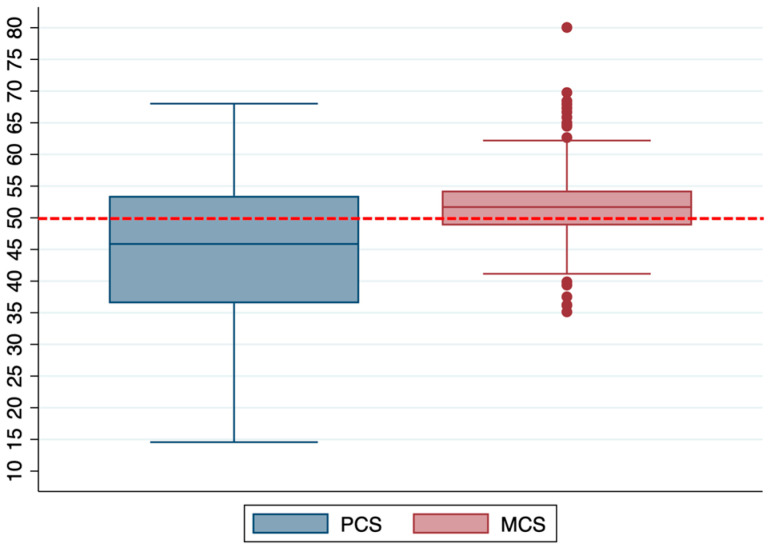
Distribution of the PCS and MCS in the entire population. Box plot shows median and interquartile ranges, while whiskers represent the outliers. The red dotted line represents the value for the normal population.

**Figure 3 jcm-12-01058-f003:**
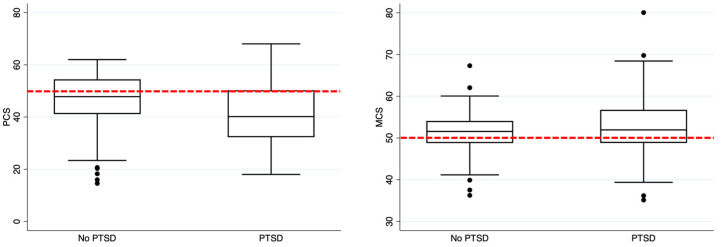
HRQoL comparison between patients without and with PTSD. In the left part of the figure, it represents the comparison between no PTSD versus PTSD patients regarding the PCS, while in the right part of the figure, the MCS comparison is shown.

**Table 1 jcm-12-01058-t001:** Baseline patients’ characteristics.

	*n* = 343
Age, median (IQR)	63 (57–70)
Men, N (%)	272 (79.3)
Weight at baseline (Kg), median (IQR)	86 (78–98)
Weight at follow-up (Kg), median (IQR)	85 (75–95)
Weight variation (Kg) median (IQR)	−2.5 (0;−7)
Cardiovascular disease, N (%)	
Arterial hypertension	147 (42.9)
Chronic artery disease	9 (2.6)
Other	11 (3.2)
Pulmonary disease, N (%)	
COPD	21 (6.1)
Pulmonary fibrosis	5 (1.5)
Asthma	10 (2.9)
Emphysema	2 (0.6)
Other	15 (4.4)
Kidney disease, N (%)	
Chronic renal failure	12 (3.5)
Other	5 (1.5)
Liver disease, N (%)	
Cirrhosis	3 (0.9)
Other	7 (2.0)
Diabetes, N (%)	49 (14.2)
Level of education, N (%)	
Low	157 (46)
High	186 (54)
Marital status, N (%)	
Single	36 (10.5)
Married/cohabiting	273 (79.6)
Separated/divorced	20 (5.8)
Widowed	14 (4.1)
Living alone, N (%)	58 (17)
Employment, N (%)	
Jobless	28 (8.2)
Active	159 (46.3)
Retired	156 (45.5)

Values are shown as medians and interquartile ranges (IQRs) or frequencies and percentages. Legend: COPD = chronic obstructive pulmonary disease.

**Table 2 jcm-12-01058-t002:** Main hospital and ICU data.

	*n* = 343
APACHE II score at ICU admission, median (IQR)	11 (8–14)
Lung injury score at ICU admission, median (IQR)	3 (2.3–3)
Mechanical ventilation, N (%)	343 (100)
Noninvasive	32 (9)
Endotracheal intubation	311 (91)
MV duration (days), median (IQR)	10 (2–20)
NMB administration	111 (32.3)
Duration of NMB (hours), median (IQR)	96 (36–160)
Steroid administration	194 (56.5)
Duration of steroid treatment (days), median (IQR)	10 (7–13)
RRT in ICU, N (%)	23 (6.7)
Tracheostomy in ICU, N (%)	87 (25.4)
Day of ICU stay at tracheostomy execution, median (IQR)	11 (6–17)
Length of hospital stay before ICU admission, (days) median (IQR)	2 (1–4)
LOS_ICU_, (days) median (IQR)	13 (6–25)
LOS_HOSP_, (days) median (IQR)	32 (19–47)
Discharge ward after ICU, N (%)	
Medical	83 (24.1)
Pulmonology	82 (24)
HDU	91 (26.5)
Other	56 (16.3)
Not specified	31 (9)

Values are shown as medians and interquartile ranges (IQRs) or frequencies and percentages. Legend: APACHE II score = acute physiologic assessment and chronic health evaluation II score, MV = mechanical ventilation, NMB = neuromuscular blockers, RRT = renal replacement therapy, ICU = intensive care unit, LOS_ICU_ = length of ICU stay, LOS_HOSP_ = length of hospital stay, HDU = high dependency unit.

**Table 3 jcm-12-01058-t003:** HRQoL according to the SF-36 items and comparison between patients with and without PTSD.

SF-36 Parameter	Overall (*n* = 343)	No PTSD (*n* = 234)	PTSD (*n* = 109)	*p*-Value
Physical Functioning	85 (60–95)	85 (70–95)	70 (40–90)	<0.001
Physical Role	75 (0–100)	75 (0–100)	25 (0–100)	<0.001
Emotional Role	100 (33–100)	100 (50–100)	33.3 (0–100)	<0.001
Bodily Pain	77.5 (45–100)	84 (55–100)	67.5 (41–100)	0.001
General Health	55 (35–72)	60 (45–75)	45 (20–60)	<0.001
Vitality	55 (40–70)	60 (45–75)	50 (30–60)	<0.001
Social Functioning	75 (50–100)	87.5 (50–100)	55 (37.5–75)	<0.001
Mental Health	68 (52–84)	72 (60–88)	56 (44–72)	<0.001
Health change	50 (25–75)	50 (25–75)	48 (25–75)	0.041
PCS	45.9 (36.5–53.5)	47.8 (41.1–54.4)	40.2 (32.2–50.3)	<0.001
MCS	51.7 (48.8–54.3)	51.6 (48.8–54.1)	51.9 (48.8–56.8)	0.123

Values are expressed as median and interquartile ranges. Legend: PF = physical functioning, PR = physical role, RE = emotional role, BP = bodily pain, GH = general health, VT = vitality, SF = social functioning, MH = mental health, PCS = physical component summary, MCS = mental component summary.

## Data Availability

The data are available from the corresponding author upon reasonable request.

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
