# Peer review of "Quality of Life in COVID-Related ARDS Patients One Year after Intensive Care Discharge (Odissea Study): A Multicenter Observational Study"

_jcm, 2023, doi:10.3390/jcm12031058_

Round 1

Reviewer 1 Report

Thank you for the opportunity to read the manuscript entitled “QUALITY OF LIFE IN COVID-RELATED ARDS PATIENTS 1 2 YEAR AFTER INTENSIVE CARE DISCHARGE (ODISSEA 3 STUDY): A MULTICENTER OBSERVATIONAL STUDY.”

This paper is well written and shows interesting results.

I have few comments

-          In fig 1, could you specify “other” in the excluded patients. Patients lost to follow-up were included in “other” ?

-          In the Results section, the last paragraph on the multivariate linear regression analysis is not very readable, could you lighten this paragraph by using tables or figures.

Author Response

We appreciate our manuscript's thorough review and the helpful comments the reviewers provided. We have undertaken a carefully comprehensive overview of these comments. All the changes are marked in red in the revised manuscript. Please find a copy of each of the reviewers' comments, along with a point-by-point response in bold font:

REVIEWER 1:

Thank you for the opportunity to read the manuscript entitled

"QUALITY OF LIFE IN COVID-RELATED ARDS PATIENTS 1 2 YEAR AFTER INTENSIVE CARE DISCHARGE (ODISSEA 3 STUDY): A MULTICENTER OBSERVATIONAL STUDY."

This paper is well written and shows interesting results. I have few comments. In fig 1, could you specify "other" in the excluded patients. Patients lost to follow-up were included in "other"? In the Results section, the last paragraph on the multivariate linear regression analysis is not very readable, could you lighten this paragraph by using tables or figures.

In fig 1, could you specify "other" in the excluded patients. Patients lost to follow-up were included in "other"?

Reply: You are right, and we are grateful to the reviewer for having pulled this off. Among 90 patients included, 44 were excluded because they provided incomplete questionnaires, while 46 were lost at follow-up, so they were not considered in the analysis. Accordingly, we modified figure 1.

In the Results section, the last paragraph on the multivariate linear regression analysis is not very readable, could you lighten this paragraph by using tables or figures.

Reply: We agree with the review, and to address this comment, we modified the paragraph by placing the results into the table in supplementary material 3.

Reviewer 2 Report

A study of interest with good design and clinically signifocant results, though not of a high novelty. The research group proved that HRQoL in C-ARDS patients was reduced regarding 69 PCS, while median MCS was slightly above normal value. Some risk factors for lower HRQoL have 70 been identified, PTSD presence being one of them. Further research is warranted to better identify 71 possible factors affecting HRQoL in C-ARDS.

The fact thet C-ARDS patients suffered from single organ failure should be better described and explained based on pathophusiology of COVID and MODS.

Reference list should include more relevant studies.

Author Response

REVIEWER 2: A study of interest with good design and clinically significant results, though not of a high novelty. The research group proved that HRQoL in C-ARDS patients was reduced regarding 69 PCS, while median MCS was slightly above normal value. Some risk factors for lower HRQoL have 70 been identified, PTSD presence being one of them. Further research is warranted to better identify 71 possible factors affecting HRQoL in C-ARDS.

The fact that C-ARDS patients suffered from single organ failure should be better described and explained based on pathophusiology of COVID and MODS. Reference list should include more relevant studies.

Reply: Thank you very much for your helpful suggestions. Based on your remarks and the available pathophysiology of COVID and MODS, we analysed more in detail that C-ARDS patients suffered from single organ failure. We also enclosed the suggested references:

1) Multiple organ dysfunction in SARS-CoV-2: MODS-CoV-2. Expert Rev Respir Med. 2020 Sep;14(9):865-868. doi: 10.1080/17476348.2020.1778470. Epub 2020 Jun 22. PMID: 32567404; PMCID: PMC7441756.

2) Long-COVID Syndrome and the Cardiovascular System: A Review of Neurocardiologic Effects on Multiple Systems. Curr Cardiol Rep. 2022 Nov;24(11):1711-1726. doi: 10.1007/s11886-022-01786-2. Epub 2022 Sep 30. PMID: 36178611; PMCID: PMC9524329.

3) The mechanism of multiple organ dysfunction syndrome in patients with COVID-19. J Med Virol. 2022 May;94(5):1886-1892. doi: 10.1002/jmv.27627. Epub 2022 Feb 8. PMID: 35088424; PMCID: PMC9015222.

4) Viral Endothelial Dysfunction: A Unifying Mechanism for COVID-19. Mayo Clin Proc. 2021 Dec;96(12):3099-3108. doi: 10.1016/j.mayocp.2021.06.027. Epub 2021 Aug 19. PMID: 34863398; PMCID: PMC8373818.

5) The clinical characters and prognosis of COVID-19 patients with multiple organ dysfunction. Medicine (Baltimore). 2021 Oct 15;100(41):e27400. doi: 10.1097/MD.0000000000027400. PMID: 34731113; PMCID: PMC8519259.

6) Physical and psychological impairment in survivors of acute respiratory distress syndrome: a systematic review and meta-analysis. Br J Anaesth. 2022 Nov;129(5):801-814. doi: 10.1016/j.bja.2022.08.013. Epub 2022 Aug 26. PMID: 36116979.

In closing, we hope our modification manuscript addressed all the issues raised in your informative and thorough review of our paper.

We would be glad to make further changes upon request. On behalf of the other authors, we extend my gratitude for your time and assistance with our review. We look forward to hearing your response.

Kind regards.